# Exploring the Pathogenesis of Spondylarthritis beyond HLA-B27: A Descriptive Review

**DOI:** 10.3390/ijms25116081

**Published:** 2024-05-31

**Authors:** Ruxandra-Elena Nagit, Elena Rezus, Petru Cianga

**Affiliations:** 1Immunology Department, “Grigore T. Popa” University of Medicine and Pharmacy, 700115 Iași, Romania; ruxandra_nagit@yahoo.com; 2Rheumatology Department, “Grigore T. Popa” University of Medicine and Pharmacy, 700115 Iași, Romania; elena.rezus@umfiasi.ro; 3Clinical Rehabilitation Hospital, 700661 Iași, Romania; 4Immunology Laboratory, “St. Spiridon” Clinical Hospital, 700111 Iași, Romania

**Keywords:** spondylarthritis, HLA-B27, pathogenesis, heritability, microbiota

## Abstract

Spondylarthritis (SpA) is a chronic inflammatory condition that encompasses damage to the axial or peripheral skeleton, accompanied by specific extra-articular symptoms. Within this group, Ankylosing Spondylitis stands out as the hallmark member. Although the heritability of Ankylosing Spondylitis is estimated to be over 95%, only a portion of the heritability has been explained, with HLA-B27 accounting for 20.1% of it; therefore, ongoing research endeavors are currently concentrated on investigating the potential participation of different entities in the development of the disease. Genome-wide association studies have led to significant advances in our understanding of the genetics of SpA. In this descriptive review, we delve into the pathogenesis of Spondylarthritis beyond HLA-B27. We summarize the latest research on the potential participation of various entities in the development of the disease, including other genetic loci, immune dysregulation, microbiota, and environmental factors. The multifactorial nature of SpA and the complex interplay of genetic, immunological, and environmental factors are being increasingly recognized; therefore, it is of paramount importance to consider a holistic approach to comprehend the pathogenesis of SpA in order to identify novel therapeutic targets.

## 1. Introduction

Spondylarthritis (SpA) constitutes a group of chronic inflammatory conditions linked by shared genetic background, pathological mechanisms, and clinical expression. These conditions involve damage to the axial or peripheral skeleton, often accompanied by specific extra-articular symptoms such as anterior uveitis, Psoriasis, and inflammatory bowel disease [1,2,3,4,5]. The clinical–radiographic presentation of SpA includes several subtypes, such as Ankylosing Spondylitis (AS), Psoriatic Arthritis (PsA), arthritis associated with inflammatory bowel diseases, reactive arthritis (ReA), and undifferentiated Spondylarthritis (uSpA) [6,7]. SpA is classified based on the predominant osteoarticular involvement in axial (axSpA) and peripheral Spondylarthritis [7,8,9]. Regarding axSpA, two forms of the condition are currently recognized—non-radiographic axial SpA (nr-axSpA) and Ankylosing Spondylitis (AS) [6]. These forms may be viewed as progressive stages of the same clinical entity, distinguished by definite sacroiliitis on conventional radiography, as defined by the modified New York criteria [10].

Ankylosing Spondylitis is the hallmark of axial Spondylarthritis, which primarily involves the axial skeleton, leading to chronic inflammatory back pain, spinal stiffness, and limited mobility. If left unaddressed, AS can progress to severe spinal deformities, such as vertebrae fusion and functional impairment [1]. On the other hand, peripheral involvement (pSpA) presents with joint inflammation affecting the extremities, potentially leading to dactylitis or enthesitis [7]. In addition to the previously mentioned extra-articular manifestations, patients diagnosed with SpA may also experience a range of cardiac manifestations, including aortitis and conduction impairments [11].

### 1.1. The Prevalence and Economic Burden of Spondylarthritis

The onset of axSpA typically occurs around the age of 26 [12], a pivotal time for personal and social development and career growth. This not only places a physical strain on patients but also a significant psychological one [13], mainly when there is a prolonged delay between symptom onset and diagnosis. Despite progress in the diagnosis of SpA, delays in detection continue to persist and vary considerably across different regions of the world, with reported median delays ranging from 1 to 8 years, as comprehensively reviewed by Hay et al. [14].

The prevalence of SpA varies widely across different geographic regions. A meta-analysis conducted in 2016 revealed that the prevalence of SpA ranged from 0.20% in Southeast Asia to 1.61% in Northern Arctic communities. Similarly, the prevalence of AS ranged from 0.02% in sub-Saharan Africa to 0.35% in Northern Arctic communities, while the prevalence of PsA ranged from 0.01% in the Middle East to 0.19% in Europe. The prevalence of undifferentiated SpA varies significantly as well, ranging from 0.0 to 0.7% [15].

Despite these variations, a growing body of evidence from various regions and healthcare systems highlights the substantial economic impact of SpA. The burden of SpA encompasses healthcare resource utilization, which comprises direct costs such as doctor visits, diagnostic workups, hospital care, medications, and out-of-pocket expenses. In addition, indirect costs such as work disability and household work contribute significantly to the disease burden [16]. As the disease primarily affects young people in the productive age range, indirect costs stemming from work absenteeism and decreased productivity due to functional limitations in patients play a pivotal role [17,18,19]. A Brazilian study revealed that nearly 59% of the study group was retired or on leave from work as a result of the disease [18].

The annual direct costs per patient of SpA vary across different countries and are influenced by several factors, including longer disease duration, lower education levels, poorer physical function, and higher disease activity [20]. The economic burden of axSpA per AS patient per year ranges from EUR 5190 in Germany [21] to USD 23,183.56 in Brazil [18]. Moreover, data from Poland has shown an increase in healthcare expenses related to AS in recent years—from USD 6.3 million in 2008 to USD 21,900 million in 2013, which might be partly related to the increased number of patients who are treated with biological therapies [22,23] or to increasing availability and utilization of MRI as a diagnostic tool in SpA [18].

Delayed diagnosis of axSpA has been linked to increased healthcare resource utilization and costs, heightened social disabilities, and work disability. Already comprehensively reviewed by Yi et al., multiple studies proved that longer diagnosis delay was associated with higher costs related to doctor visits and specialist services, unnecessary spinal surgery and treatments, and also with indirect costs related to loss of employment due to a greater likelihood of work disability [24]. Studies have also highlighted the financial repercussions of delayed diagnosis of axSpA, with data from Italy revealing substantial expenses of approximately EUR 153,000 incurred in the 3 years prior to a confirmed SpA diagnosis [25]. On the other hand, a correct diagnosis of axSpA results in substantial health and cost benefits for patients and society; for each correctly diagnosed axSpA patient, up to EUR 60,000 could be gained/saved [26].

### 1.2. HLA-B27 and Spondylarthritis

It is widely recognized that Spondylarthritis has a strong familial component, with the heritability of AS estimated to be over 95% [27]. Despite this, only a portion of the heritability—around 24.4%—has been explained, with HLA-B27 accounting for 20.1% and other loci accounting for 4.3% [28]. Since 1973, the role of the Class I Major Histocompatibility Complex (MHC), in the form of HLA-B27, has been established by two separate research teams led by Brewerton and Schlosstein [29,30].

The MHC class I molecules play a crucial role in presenting intracellular peptides to CD8+ T lymphocytes. Unlike other MHC class I molecules, HLA-B27 presents a series of peculiarities that can cause it to misfold, form homodimers, and present improperly bound peptides, leading to the development of Spondylarthritis [31]. The presence of an unpaired cysteine residue at position 67 (Cys67) in the B27 molecule enables the formation of homodimers through the creation of disulfide bridges between its heavy chain amino acids [32]. Moreover, HLA-B27 has the ability to bind longer peptides of up to 33 amino acids, which is much higher than the typical 8–11 amino acids of other MHC class I molecules. As a result of this unique feature, HLA-B27 has a much looser folded conformation, making it more predisposed to misfolding and homodimer formation. The folding of the heavy chain of HLA-B27 appears to be slower than many other class I molecules, leading to a tendency to unfold during its synthesis in the endoplasmic reticulum (ER), even under normal peptide and β2m binding conditions [33]. Furthermore, HLA-B27 is less dependent on Tapasin for surface expression in the ER, which results in the release of HLA-B27 complexes loaded with improperly bound peptides. These complexes may dissociate upon reaching the cell surface, causing β2m to dissociate as well, leaving the MHC “empty” [31].

To this day, it remains the most extensively researched and potent genetic association with the disease. Between 74 and 95% of AS patients are HLA-B27-positive, which makes it a crucial diagnostic marker in a suggestive clinical setting [34,35,36]. HLA-B27 positivity not only aids in the diagnosis of AS but also has prognostic implications, as HLA-B27-positive individuals often present with specific clinical features, such as earlier disease onset, more severe axial inflammation, and greater radiographic damage of the sacroiliac joints [37]. Conversely, a negative HLA-B27 status is a leading contributor to delayed diagnosis. Therefore, ongoing research endeavors are currently concentrated on delving into the potential participation of other entities in the development of the disease [38]. In this regard, several reviews have been published on the pathogenesis of Spondylarthritis, featuring remarkable contributions from distinguished researchers like Reveille et al. [39], Wordsworth et al. [40], Brown et al. [41], Busch et al. [42], Breban [43], and Bowness [44], among others. Nonetheless, in light of the latest research developments, we considered that a fresh perspective would be helpful not only in terms of presenting comprehensive data but also in terms of structuring the information.

## 2. Various Genetic Factors, Other Than HLA-B*27, Are Involved in the Development of Spondylarthritis

The high prevalence of Ankylosing Spondylitis among individuals carrying the HLA-B*27 allele is a well-established observation, yet the exact molecular mechanism responsible for this association remains elusive. One of the earliest theories is that autoimmunity is caused by the similarity between a pathogen and self-epitopes due to the unique antigen presentation function of HLA-B27 [43]. More recent theories suggest a direct pro-inflammatory role of HLA-B27, which induces Th17 axis activation following misfolding of the HLA-B27 heavy chain or formation of homodimers by this chain on the cell surface [43].

Over the last years, extensive research efforts have been undertaken to identify genetic associations in SpA beyond HLA-B*27. The study of these associations has primarily been carried out through direct genotyping and imputation-based methods. One commonly used approach is genome-wide association studies (GWAS), which involves searching for correlations between genome polymorphisms (usually single nucleotide polymorphisms, or SNPs) and a phenotype of interest, albeit at very high costs [45,46]. An alternative approach, the Immunochip, is an Illumina Infinium SNP microarray chip with a dense coverage of the MHC and killer immunoglobulin-like receptor (KIR) loci, produced for a fraction of the cost of GWAS chips. It covers a variety of autoimmune and seronegative conditions, including Ankylosing Spondylitis and related illnesses such as Ulcerative colitis, Crohn’s disease, and Psoriasis, along with Rheumatoid Arthritis, Systemic Lupus Erythematosus, Type I Diabetes, Autoimmune Thyroiditis, Celiac Disease, and Multiple Sclerosis [45].

The majority of genome-wide association studies aiming to investigate Ankylosing Spondylitis have been carried out on individuals of Caucasian descent. The Wellcome Trust Case Control Consortium (WTCCC) and Australo-Anglo-American Spondylitis Consortium (TASC) conducted the first AS GWAS in 2007, analyzing 14,500 nonsynonymous SNPs in 1000 AS patients and 1500 healthy controls among British and North American Caucasians [47]. In 2010, TASC performed a comprehensive GWAS on a larger cohort of 2053 AS patients and 5140 controls of European ancestry [48]. A larger AS GWAS was published in 2011 by TASC and WCCC2, which involved a cohort of 3023 AS patients and 9141 healthy controls of European descent. Concurrently, Lin and colleagues conducted the only GWAS to date in the Han Chinese population, involving a total of 3937 individuals with Ankylosing Spondylitis and 8177 controls [49]. In 2013, the International Genetics of Ankylosing Spondylitis Consortium (IGAS) performed a case-control AS association study on the Immunochip, which involved 10,619 AS cases and 15,145 controls in populations of European, East Asian, and Latin American ancestry [28].

Two further studies were conducted in 2015 and 2016, respectively. The first study, conducted by Robinson, employed the Illumina Exomechip to analyze AS associations in a cohort comprising 5040 AS patients and 21,133 healthy controls of European descent [50]; the second study, conducted by Ellinghaus, involved 8726 AS cases and 34,213 healthy controls and employed an AS case-control association study on the Immunochip [51]. The only genome-wide association study in the Middle East to date was executed by Li and colleagues in 2019. The study involved 1001 Turkish patients diagnosed with Ankylosing Spondylitis (AS) and 1011 Turkish controls, as well as 479 Iranian AS patients and 830 Iranian controls [52]. In 2020, Wang et al. conducted a genotyping study using the Illumina Immunochip on a total of 1637 Chinese, Taiwanese, and Korean AS cases and 1589 ethnically matched controls [53]. In 2015, Cortes et al.’s study of European-descent subjects involved genotyping 7264 MHC SNPs in 22,647 AS cases and controls, also of European descent, using the Illumina Immunochip microarray. They later used statistical imputation methods to impute SNPs, classical HLA alleles, and amino-acid residues within HLA proteins and tested these for association with AS [54]. The same group reunited in 2018 to examine the relative contributions of HLA class I and class II alleles in three ethnic groups: 1948 whites of European ancestry, 442 Han Chinese, and 67 African Americans. They used Single-Stranded Conformational Polymorphism (SSCP) typing of HLA-A, HLA-B, and HLA-C alleles and used statistical analysis to determine overall disease associations, conditioning on the presence of HLA-B27 [55].

According to the aforementioned studies, the latest available data acknowledges the role played by both intra- and extra-HLA genes in determining the onset and progression of Spondylarthritis. While HLA associations will be discussed in detail later, subsequent sections aim to provide an overview of the most frequent molecules outside HLA.

As of today, over 114 non-MHC variants that have reached genome-wide significance have been identified. Table 1 presents a non-exhaustive timeline of the discoveries in question. Some of these variants are involved in antigen presentation (aminopeptidases), some in Th17 responses (IL6R, IL23R, TYK2, and STAT3), and others in macrophages and T-cell responses (IL7R, CSF2, RUNX3, and GPR65) [28,47,48,49,50,51,53]. Further notable associations have also been described with genes like ANTXR2, KIF21B, LTBR-TNFRSF1A, PTGER4, TBKBP, FCGR2A, UBE2E3, GPR35, BACH2, ZMIZ1, SH2B3, GPR65, SULT1A1, NOS2, ICOSLG, ITLN1, CTLA4, CMC1, NPM1P17, NFKB1, CDKAL1, FGFR10P, ACTA2, PPP2R3C, CORO1A, ERN1, PTPN2, and FAM118A, as well as intergenic regions of 2p15, 21q22, 6p22, 7p21, 11q24, and 16p11 [28,47,48,49,50,51,53].

### 2.1. The Involvement of the IL-23R Pathway in AS Pathogenesis

The gene IL-23R encodes the receptor for IL-23, a cytokine that activates various proinflammatory cells, including Th17 T cells, gamma-delta T cells, NK cells, Mast cells, and Paneth cells [57]. The IL-23R locus has been linked to several autoimmune diseases such as inflammatory bowel disease [58], Behçet’s disease [59], Psoriasis [60], and Ankylosing Spondylitis. A number of IL-23 R polymorphisms have been found to be significantly associated with Ankylosing Spondylitis, with the most frequent ones listed in Table 2. Among these, rs11209026, rs11465804, rs11209032, rs10889677, rs1004819, rs7517847, and rs2201841 have been extensively studied in the Caucasian population [47,48,56,61,62,63,64,65]. However, a considerable degree of variation in the association of IL-23R with AS has been observed across different ancestral populations. For instance, the IL-23R SNPs linked to AS in European descent populations do not seem to be associated with AS in Han Chinese populations [66,67,68]; this discrepancy has been partially attributed to the absence of the key European-associated variant, rs11209026, in East Asians. Instead, a different variant, rs76418789 (G149R), has been identified as AS-associated in East Asian populations [69].

Additionally, various molecules that participate in the signaling pathway of IL-23R, including but not limited to CARD9, EOMES, IL1R1/R2, IL6R, IL7R, IL12B, IL27, RUNX3, NKX-2, PTGER4, TBX21, and TYK2, have been identified as having an association with Ankylosing Spondylitis [47,48,56].

### 2.2. Role of Aminopeptidases—A HLA-B27-Dependent Mechanism and Beyond

The proper stabilization and membrane expression of classical MHC molecules require the loading and binding of a peptide in the MHC cleft. In the case of MHC-I molecules, such as HLA-B27, the antigen direct presentation pathway involves proteasomal degradation in the cytosol, the Transporter Associated Peptide (TAP)-mediated peptide transport in the endoplasmic reticulum (ER), followed by the uptake and presentation of MHC-I peptide to CD8+ T cells. However, some partially processed peptides that are too long for the MHC-I cleft and cannot be accommodated in its groove require further trimming by enzymes such as ERAP (endoplasmic reticulum aminopeptidase associated with antigen processing), LNPEP, or NPEPPS, as reviewed by Paladini et al. [72].

It was shown that peptide processing and presentation by HLA-B molecules play a crucial role in the development of Ankylosing Spondylitis [56]. The association of Ankylosing Spondylitis with aminopeptidases located either inside the endoplasmic reticulum (ER)—such as ERAP1 [47,48,54,56,64,73,74,75,76,77,78,79,80,81,82,83,84], ERAP2 [28,50], and LNPEP [28,47]—or in the cytoplasm—specifically NPEPPS [28,85]—highlights the importance of this process. Among these, the presence of ERAP1 is currently recognized as the second-most-important genetic determinant of Ankylosing Spondylitis after HLA-B27; however, the exact mechanism that confers this increased risk remains uncertain [86].

ERAP1 and ERAP2 differ in their substrate and cleavage specificity, mainly determined by the N-terminal residues of the substrate. ERAP1 prefers hydrophobic residues, except for proline, while ERAP2 prefers basic residues. ERAP1 is more efficient in cleaving larger substrates, while ERAP2 optimally cleaves peptides consisting of 7–8 residues. ERAP2 also plays a role in potentiating ERAP1-mediated processing [87].

ERAP1

ERAP1 is a highly polymorphic enzyme that comprises ten haplotypes, Hap1–Hap10, determined by various single nucleotide polymorphisms (SNPs). The activity of the natural variants of ERAP1 depends on the specific combination of these residues [88].

The association between AS and ERAP1 variants was first identified in a genome-wide association study conducted by the Wellcome Trust Case Control Consortium and the Austro-Anglo-American Spondylitis Consortium in 2007. This study revealed that ERAP1 contributes up to 26% of the overall risk of developing AS in the North American population [47]. The association between AS and ERAP1 variants has been subsequently replicated in other genome-wide association studies and several case-control independent studies in various populations. To date, more than 33 polymorphisms have been identified as potential genetic determinants of AS [82], with Table 3 listing some of the most frequently observed polymorphisms. Three of these genetic variations have gained significant attention due to their highest frequency in diverse populations. The first variation, rs30187 (K528R), results in the substitution of Arg for Lys at position 528 and has been linked to a reduction in the activity of the ERAP1 amino-peptidase enzyme towards a peptide substrate [56]. The second variation, rs10050860 (D575N), affects indirectly the specificity or enzymatic activity by altering the conformational change between open and closed forms, as it is located at the domain junction [77]. The third variation, rs27044 (Q730E), impacts the specificity of substrate sequence or length due to its exposure on the inner surface of the C-terminal cavity [77]. Furthermore, the combination of these three main genetic variations (rs30187, rs17482078, and rs10050860) determines three major haplotypes associated with SpA—the “protective” haplotype T/T/C, the “neutral” haplotype C/C/C, and the “susceptibility” haplotype C/C/T [77].

The initial data indicated that the association between Ankylosing Spondylitis and ERAP1, specifically the rs30187 variant, was limited to cases that tested positive for HLA-B27 [56]. This postulation was reinforced by subsequent research by Cortes et al., who found an epistatic interaction between ERAP1 SNP rs30187 and HLA-B*27 in AS [54]. Additionally, the same study demonstrated a similar interaction between ERAP1 and the HLA-B*40 allele.

The co-occurrence of ERAP1 and HLA-B27 has been attributed to a pathogenetic mechanism that involves the alteration of the peptide repertoire presented to HLA-B27. This alteration may cause compromised recognition by cytotoxic T- or NK-cells [90]. The disease-associated ERAP1 variants have shown a faster rate of peptide trimming than the protective ERAP1 variants [56]. In addition, ERAP1 has been shown to influence the folding ability of MHC class I molecules. A recent study conducted on ERAP1-knockout rats has revealed that the deficiency of ERAP1 resulted in a reduction in HLA-B27 misfolding and an improvement in folding. This, in turn, provided partial protection against SpA [91]. Furthermore, a decrease in ERAP1 activity has also been shown to decrease the formation of homodimers by free HLA-B27 heavy chains not only on cell lines like the human lymphoblastoid C1R B cell line or the HeLa epithelial cell line but also on AS patients’ monocytes [92].

However, the involvement of ERAP1 in the pathogenesis of Ankylosing Spondylitis extends beyond its association with the HLA-B27 molecule. Specific ERAP1 polymorphisms have been associated with the modulation of the interleukin IL-17/IL-23 inflammatory axis in Iranian AS patients [93]. Further data demonstrate associations of ERAP1 not only with Ankylosing Spondylitis but also with other AS-related pathologies such as Psoriasis and Behcet’s disease. This observation can be attributed to the fact that these diseases share common genetic susceptibility factors with AS [94,95]. ERAP1 has been found to interact with HLA-C*06:02 in the context of Psoriasis in a similar manner as it does with HLA-B27 in Ankylosing Spondylitis [94]. This might be attributed to the partial overlap in the peptidome of the molecules coded by HLA-B*27 and HLA-C*06:02, which share identical peptide-anchoring residues at positions P2 and P9 [96]. Another mechanism through which ERAP1 may interact with HLA-C06:02 in Psoriasis involves the enzymatic trimming of NH2-terminal elongated peptide precursors to the appropriate length, thereby facilitating their presentation by HLA-C06:02. This process ultimately results in the generation of the causative melanocyte autoantigen [97]. Pathogenic interactions have also been shown between ERAP1 and the HLA-B51 protein in Behcet’s disease [95].

The data mentioned above are primarily derived from the Caucasian population. It is noteworthy that the ERAP1 variants commonly referred to, such as rs30187, rs27037, rs27980, rs27434, rs27582, and rs7711564, have been found to exhibit no correlation with AS in the Chinese Han population [66,98].

ERAP2

In the endoplasmic reticulum, ERAP1 and ERAP2 enzymes can form heterodimers that exhibit enhanced substrate processing capabilities compared to independent trimming activities [99]. The physical interaction between the enzymes induces allosteric changes in ERAP1, leading to increased substrate-binding affinity and enzymatic activity [100].

ERAP2 is also known to be highly polymorphic, similar to ERAP1; however, unlike ERAP1, 25% of the individuals may experience a lack of ERAP2 expression due to the effects of these polymorphisms [101]. One of the most frequent SNPs is the rs2549782, which encodes the K392N substitution. The K392 allele causes increased levels of ERAP2, while the N392 allele promotes RNA decay, thus impacting protein expression, which results in a change in ERAP2 catalytic activity [102]. The allele coding for N392 is in tight linkage disequilibrium with the rs2248374 variant. The rs2248374 leads to the transcription of a truncated ERAP2 mRNA, which undergoes degradation, resulting in a complete absence of the ERAP2 protein [101].

ERAP2, specifically in the form of rs2910686 variant, was identified in 2013 as an independent risk factor for developing Ankylosing Spondylitis in HLA-B27-negative patients [28]. Two years later, it was found that the expression of ERAP2, particularly in the form of rs2248374 or rs2549782, is associated with an increased risk of AS in both HLA-B27-positive and HLA-B27-negative patients [50]. Martin-Esteban et al. demonstrated that ERAP2 (rs2549782) could directly influence the peptidome bound to B27:05 by cleaving some ligands and enhancing the activity of ERAP1, causing a shift from an optimal “ligandome” to a “mis-peptidome”. This shift can significantly impact the adaptive CD8+ T-cell and the innate NK-cell immune responses [103]. Based on extrapolation, it is plausible that this effect on the peptidome of other MHC class I molecules determines its involvement in the pathogenesis of HLA-B27-negative Spondylarthritis [103]. Subsequently, a study in Romania has found that the rs2248374 variant is significantly associated with Psoriatic Arthritis among HLA-B27-negative individuals [89].

In 2018, Paladini et al. presented evidence for a coordinated quantitative regulation of ERAP genes, namely, ERAP1 and ERAP2. The authors identified a single nucleotide polymorphism (rs75862629), located in the promoter region of the ERAP2 gene, that displayed functional effects on the transcriptional regulation of both ERAP1 and ERAP2. Specifically, the presence of a G nucleotide instead of an A at this locus resulted in the upregulation of ERAP1 mRNA and concomitant downregulation of ERAP2 mRNA expression [104]; moreover, the authors demonstrated that this SNP conferred protection against AS in HLA-B27-positive individuals in Sardinia [105].

MICA

MICA (MHC class I polypeptide-related sequence A) plays a role in generating the NK cell-dependent immune response. Also coded by a polymorphic gene, it has further been linked to an increased risk of Ankylosing Spondylitis. Research conducted in 2005 on a group of Algerian patients has shown that a MICA-129 dimorphism may be associated with juvenile AS, regardless of the patient’s HLA-B27 status [106].

Later, MICA*019 was described as a significant risk allele for AS in Chinese patients, while MICA*007:01 showed a similar risk in both Caucasian and Han Chinese populations in a large genotyping study conducted by Zhou [107]. Building upon previous reports of strong linkage disequilibrium (LD) between HLA-B and MICA SNPs [28], Cortes et al. did not find an association between the MICA*007 allele and susceptibility to AS; however, the authors have identified a strong linkage disequilibrium between HLA-B*27 and MICA*007, possibly accounting for the previously reported associations with this allele [108].

### 2.3. The Involvement of Non-B27 MHC Class I

HLA-B7 CREG

The HLA-B7 antigen and the B7 CREG (“cross-reactive group”): B7, B22, and B42 were some of the first “alternative MHC alleles” proposed. In 1977, Arnett and his team observed that these alleles were more frequent in patients with reactive arthritis and idiopathic sacroiliitis [109]. A few years later, in 1983, it was confirmed that B22 and B42 belonged to the B14 cross-reactive group [110].

Further research has revealed a link between the B7 antigen and AS in Black patients [111] and certain types of axial SpA [112] and undifferentiated Spondylarthritis [113] among Tunisian [113], Brazilian [114], West Indian [115] patients; however, later studies indicate a negative relationship between B7 and Ankylosing Spondylitis [54,55,116].

B14 was first described in two Spondylarthritis patients from the same family [117], and later in AS patients from West Africa [118,119,120]. Subsequently, in 2018, Reveille reported a significant association of B14 with AS [55].

HLA-B16 (B38, B39)

As early as over 40 years ago, higher frequencies of the serologically defined HLA-B16 antigen, now identified as HLA-B38 and HLA-B39 antigens, were demonstrated in patients with B27-negative AS [111,121]. More recent studies have confirmed that HLA-B*39 is associated with the disease in Japanese patients [122] while HLA-B*38 is associated with the disease in Caucasians [55,123]. Today, HLA-B*38 and HLA-B*39 are strongly associated alleles with susceptibility to Psoriatic Arthritis [124,125].

HLA-B15

The involvement of the serologically defined Bw35 and Bw62 antigens in HLA-B27-negative AS patients was first mentioned in 1984 by Wagener et al. [126]. In recent years, HLA-B15 of the Bw62 group has been specifically identified in patients with undifferentiated Spondylarthritis. A follow-up study conducted by Mielants on Belgian SpA patients who were negative for HLA-B27 but positive for HLA-Bw62 (HLA-B15) is worth noting. These patients were observed over seven years and exhibited disease manifestations that included asymmetric pauciarticular arthritis, enthesopathies, and sacroiliitis. The study found their condition had a favorable evolution compared to HLA-B27-positive patients [127]. Similar results were described in 2002 by Vargas-Alarcón in a Mexican population. HLA-B15 has been associated primarily with undifferentiated SpA, predominantly in late-onset forms of the disease and indicating a better prognosis than HLA-B27-positive patients [128]. Disease associations have also been demonstrated in groups of undifferentiated SpA patients from Tunisia [113] and Colombia [129].

HLA-B40 (B60, B61)

In 1995, HLA-B40 was first reported in patients with peripheral SpA [112]. A year later, Brown suggested that HLA-B60 might influence susceptibility to AS independently of B27 [34]. Furthermore, in HLA-B27-negative AS Taiwanese Chinese patients, the independent association of HLA-B60 and B61, the major subtypes of B40, has been demonstrated [130]. Cortes and Reveille also reported a moderate association between AS and B*40:01 and B*40:02 [55]

HLA-B51

Several authors have suggested the HLA-B*51 association but its validity is still debatable. It was first identified in Tunisian patients with reactive arthritis [113]. More recently, an occurrence was reported of HLA-B51:01-related AS in a Korean family (father and three of the five daughters) [131]; however, it is essential to note that HLA-B*51 is highly prevalent among the Korean population. Cortes et al. have only shown a moderate association between B*51:01 and AS through statistical imputation methods—findings that were not confirmed through HLA genotyping of the same subjects [54,55]. Interestingly, the HLA-B*51 allele has even been shown to be protective in patients with AS [116].

Table 4 includes the most frequently reported MHC Class I allele associations and the populations in which they have been described.

Other MHC I Alleles

Among the HLA-B group, several alleles were present in AS Caucasian patients, including B*13 [54], B*49, and B*52 [55]. HLA-B*57:03 was associated with undifferentiated Spondylarthritis in Africans [120], as well as Psoriatic Arthritis in Chinese [132]. On the other hand, B*57:01 has been found to be protective against AS in Caucasians [54,55].

Recent data indicate high frequencies of HLA-A*02:01 [54] and HLA-A*29 [55] in AS patients and HLA-A*01:01 in Chinese patients with Psoriatic Arthritis [132].

Axial Spondylarthritis has been associated with several other HLA alleles in specific populations. For instance, HLA-Cw02 was linked to axSpA in Moroccans [134], HLA-C*15:02 in Koreans [135], and HLA-C*12:02:02 in Taiwanese [136]. HLA-Cw6, HLA-C*12, HLA-C*06, and HLA-C*06:02 were also confirmed as risk factors for Psoriatic Arthritis in various populations [124,125,132].

Last but not least, Santos et al. demonstrated in 2018 an association between HLA-F*01:01:01/01:03:01 and susceptibility to AS, independent of HLA-B27 antigen presence [137].

### 2.4. The Involvement of MHC Class II

MHC Class II molecules have also been found to play a role in the pathogenesis of Spondylarthritis, with the most frequent of them being included in Table 5. In AS patients, high frequencies of HLA-DR4 have been identified, especially in those with additional peripheral manifestations [138]. DR1 has been described in cohorts of British AS patients [139] and uSpA Mexican patients [128]. An association of the DRB1*04 allele was also observed in Tunisian patients with reactive arthritis who were B27-negative as well as in Colombian axSpA patients [129]. The latter group also showed high frequencies of DRB1*01 [129]. In the Chinese Han Population, HLA-B27-negative women carrying the alleles A*32:01, C*08:01, and DRB1*04:05 had a higher risk of developing AS than women without these alleles [133]. Furthermore, Reveille et al. identified high frequencies of HLA-DRB1*11 and HLA-DPB1*03:01 [55] in AS patients.

## 3. The Complex Relationship between Gut Microbiome Dysbiosis and Spondylarthritis Development

The disturbance of intestinal microbiota was found to play a significant role in developing Spondylarthritis. Despite numerous studies in the last decade aiming to validate this hypothesis, a consensus has not yet been reached on the matter of whether dysbiosis contributes to the pathogenesis of Spondylarthritis or it is simply an extra-articular manifestation of systemic inflammation [140].

While only 10% of the patients with axSpA have an evident inflammatory bowel disease, more than half of them experience subclinical changes in the form of microscopic inflammatory lesions of the intestinal mucosa. The main culprits are impairments of the intestinal vascular barrier and the presence of adhesive and invasive bacteria. There is even a correlation between these lesions and the disease activity. The causality of these lesions is still controversial, except for reactive arthritis, where enthesitis and synovitis occur following a remote infection, usually at the genitourinary or gastrointestinal level, with Campylobacter, Salmonella, Yersinia, or Shigella [141].

It is possible that a shared genetic predisposition towards inflammatory disorders, characterized by polymorphism in genes that code for cytokines, their receptors, or effectors involved in their signaling pathways (such as those of IL-23R, CARD9, EOMES, STAT3, IL-1R1/R2, IL-6R, IL-7R, IL-12B, IL-27, RUNX3, NKX-2, PTGER4, TBX21, and TYK2), could explain the observed association. Additionally, the current evidence suggests that genetic determinants play a role in shaping the microbiome’s composition [142].

The microbiota refers to the collection of microorganisms in a commensal, symbiotic, or pathogenic relationship with multicellular organisms. Although often used interchangeably, the term “microbiome” encompasses the entire ecosystem of these microorganisms, including their genes and environment. In humans, the microbiota comprises bacteria, fungi, archaea, and viruses, primarily in the skin, oral and nasal cavities, and digestive and genitourinary tracts. The microbial population in the human gut is dominated by members of two bacterial phyla—*Bacteroidetes* and *Firmicutes*—and a member of the Archaea—*Methanobrevibacter smithii* [143]. *Actinobacteria* and *Proteobacteria* comprise 10% of the microbiota’s relative abundance [144].

Numerous studies have been conducted to investigate the microbiota composition of patients with Spondylarthritis, with the aim of establishing a potential correlation between the two; however, the results of these studies have been inconclusive, if not contradictory, as indicated by the data presented in Table 6, Table 7 and Table 8.

### HLA-B27 and Its Potential Role in Intestinal Microbial Dysbiosis

The presence of HLA-B27 has been suggested to influence the composition of the commensal flora. For instance, Breban et al. found that people positive for HLA-B27 had higher levels of *R. mucilaginosa* and *E. lenta*, two bacteria linked to inflammatory bowel disease, and lower levels of beneficial bacteria like *Bifidobacterium* and *Odoribacter*. These findings are similar to those observed in patients with ileal Crohn’s Disease and Ulcerative Colitis [146].

However, according to some authors, there is insufficient evidence to support the claim that HLA-B27 influences the intestinal microbial community composition [156]. This is because most studies that generated such data were not designed to investigate the role of genetics in the microbiome. Additionally, studies on a diverse population have not shown a connection between HLA-B27 and disease susceptibility [140]. Furthermore, genetically unrelated individuals who share the same household seem to exhibit significant similarities in their microbiome composition [157]. The main determinants affecting the microbiome are environmental factors such as diet or drugs [156].

Nonetheless, it has been found that patients with AS and their first-degree relatives have significantly higher permeability in their small intestine compared to control groups. This suggests that these permeability issues may be genetically determined and occur before joint or intestinal symptoms manifest, even without direct influence from HLA-B27 on the composition of the intestinal microbiota [158].

An abnormally increased passage through the intestinal barrier is known as a “leaky gut.” This condition can cause bacterial products such as lipopolysaccharides (LPS), LPS binding protein, and intestinal fatty-acid binding protein (iFABP) to translocate. Once in the bloodstream, these molecules can trigger inflammation in areas of high mechanical stress. This can occur both at the osteoarticular level, causing osteitis and synovitis, and in extra-skeletal locations such as the aortic base or the pulmonary apex [141]. Another possible cause of inflammation is the activation of innate immune system actors at the intestinal level. These actors may release pro-inflammatory cytokines that enter the bloodstream and cause inflammation at the same sites [141].

## 4. Environmental Triggers of Spondyloarthropathies

Various environmental factors can affect the development, activity, and management of SpA. For instance, smoking has been widely recognized as a contributor to long-term adverse effects in people with Ankylosing Spondylitis [159]. Stressful life events and emotional stress can also trigger or sustain inflammatory arthritis. Additionally, vaccination has been moderately linked to SpA disease activity [160]. Some data suggest that childhood appendicitis may be inversely related to adult Ankylosing Spondylitis, while respiratory tract infections may increase the risk of developing adult AS [161]. Lastly, obesity appears to have a detrimental effect on the effectiveness of axSpA treatment, as evidenced by the review conducted by Zurita Prada et al. [162].

## 5. Conclusions

Spondylarthritis is a highly complex condition with significant social and economic impact. Although we have made considerable strides in understanding the factors responsible for the disease, there is still much to be learned. The pathogenesis of the disease involves complex interplay between autoimmune and autoinflammatory mechanisms.

HLA-B27 remains the most extensively researched and potent genetic association with SpA but accounts for only 20.1% of the heritability of the disease. Although many other genes, inside and outside of the HLA complex, have been linked to SpA, none of these associations have been proven strong enough for routine genetic testing. Nevertheless, in light of current and potential future findings, it is reasonable to consider early interventions for high-risk patients, such as first-degree relatives of SpA patients, to modify the risk of developing the disease. The gut microbiome is emerging as a crucial factor in SpA pathogenesis, with gut dysbiosis linked to immune dysregulation and inflammation. Identifying environmental triggers, such as microbial infections or dietary factors, is crucial for disease prevention and management. Furthermore, the complex interplay between innate and adaptive immune responses, including the involvement of cytokines and immune cell subsets, requires further investigation to unravel the mechanisms of immune dysregulation in SpA. Understanding the tissue-specific pathology driving axial versus peripheral joint manifestations is essential for developing targeted therapeutic strategies. Current biomarkers like CRP and ESR have limitations, and there is a pressing need for more sensitive and specific biomarkers reflecting disease activity and progression. Epigenetic modifications also warrant attention, as their role in SpA pathogenesis and potential as therapeutic targets remain poorly understood. By understanding the connections between the central nervous system and the immune/inflammatory response, we may develop pharmacological or behavioral treatment solutions.

Lastly, unraveling the intricate interactions between genetic susceptibility and environmental factors will pave the way for personalized medicine approaches and preventive interventions in SpA management.

## Figures and Tables

**Table 1 ijms-25-06081-t001:** Non-MHC associations in AS.

Year	Study	AS Cases/Controls	Population	Novel Data
2007	WTCCC, TASC [47]	1000/1500	British and North American Caucasians	ERAP1, IL23R
2010	TASC [48]	2053/5140	Mixed European	ANTRX2, IL1R2, KIF21B, 2p15, 21q22
2011	TASC, WCCC2 [56]	3023/9141	Mixed European	RUNX3, LTBR-TNFRSF1A, IL12B
2011	Lin [49]	3937/8177	Han Chinese	5q14.3, 12q12
2013	IGAS [28]	10,619/15,145	European, East Asian, and Latin American	IL6R, FCGR2A, UBE2E3, GPR35, BACH2, ZMIZI1, NKX2-3, SH2B3, GPR65, SULT1A1, NOS2, TYK2, ICOSLG
2015	Robinson [50]	5040/21,133	Mixed European	USP8, CDKAL1
2016	Ellinghaus [51]	8726/34,213	European and East Asian	ITLN1, CTLA4, CMC1, NPM1P17, NFKB1, CDKAL1, FGFR10P, 6p22, 7p21, ACTA2, 11q24, PPP2R3C, CORO1A, 16p11, ERN1, PTPN2 FAM118A
2019	Li [52]	1001/1011	Turkish	M694V
		479/830	Iranian	

**Table 2 ijms-25-06081-t002:** Frequent IL-23R SNPs in AS.

SNP	SpA	Ancestry
rs11209026	AS	British and North American Caucasian [47,61], Canadian [63], Spanish [62], Swedish [70], Mixed European [48]
rs11465804	AS	British and North American Caucasian [47,61], Canadian [63], Portuguese [64]
rs11209032	AS	British, Australian, and North American of European ancestry [47,56,61], Spanish [62], Canadian [63], Portuguese [64], Korean [65]
rs10889677	AS	British and North American Caucasian [47], Portuguese [64], Canadian [63], Spanish [62], Korean [65], Chinese [71]
rs10489629	AS	British and North American Caucasian [47], Canadian [63], Spanish [62]
rs1495965	AS	Portuguese [64], British and North American Caucasian [47,61], Canadian [63], Korean [65]
rs2310173	AS	Mixed European [48]
rs2201841	AS	Canadian [63]
rs1343151	AS	British and North American Caucasian [47,61], Portuguese [64], Spanish [62]
rs1004819	AS	British and North American Caucasian [47,61], Portuguese [64], Canadia [63], Korean [65]
rs10489629	AS	British and North American Caucasian [47,61], Portuguese [64], Canadian [63], Korean [65]

**Table 3 ijms-25-06081-t003:** Frequent ERAP1 SNPs and haplotypes in SpA.

SNP	SpA	Ancestry
rs30187 (K528R)	AS	British, North American, and Australian of European ancestry [47,56,82], Portuguese [64], French and Belgian [77], Polish [81], Mixed European [54], Korean [83], Taiwanese [80]
	PsA	Romanian [89]
rs27044 (Q730E)	AS	British and North American Caucasian [47,82], Portuguese [64], Polish [81], Korean [83], Taiwanese [80], Iranian [84]
rs10050860 (D575N)	AS	British and North American Caucasian [47,82], Iranian [84]
rs27434	AS	Mixed European [48], Chinese Han [73,74,79], Korean [75]
rs27037	AS	Korean [75], Taiwanese [80]
rs7711564	AS	Chinese Han [79]
rs2287987 (M349V)	AS	Polish [81]
rs27044/10050860/30187-CCT	AS	Canadian of Northern European descent [76]
rs17482078/rs10050860/rs30187-CCT	SpA	French, Belgian [77,78]
rs30187/rs27044-CC	PsA	Romanian [89]

**Table 4 ijms-25-06081-t004:** MHC class I allele associations.

MHC Class I	SpA Associations	Ancestry
*B*07*	axSpA	French [112]
	ReA	American [109]
	Idiopathic Sacroiliitis	American [109]
	AS	American Black [111]
	uSpA	Tunisian [113], Brazilian [114], West Indian [115]
B*13:02	AS	European [54]
	PsA	Chinese [132]
B*14	SpA	French [117]
	AS	European, Asian, African [55]
B*15	uSpA	Belgian [127], Mexican [128] Tunisian [113], Colombian [129]
B*22	AS	West African [118,119,120]
B*38	AS	Caucasian [55,123]
	PsA	Argentinian [124], Canadian [125]
B*39	As	Japanese [122]
	PsA	Canadia [125]
B*40	peripheral SpA	French [112]
	AS	British [34], Taiwanese Chinese [53,130], Korean [53]
B*40:01, B*40:02	AS	European, Asian, African [54,55]
B*49	AS	European, Asian, African [55]
B*51	ReA	Tunisian [113]
*B*51:01*	AS	Korean [131], European [54]
B*52	AS	Caucasian [55]
B*57:03	uSpA	African [120]
	PsA	Chinese [132]
A*02:01	AS	European [54]
A*29	AS	European, Asian, African [55]
A*01:01	PsA	Chinese [132]
A*32:01	AS	Chinese Han [133]
C*08:01	AS	Chinese Han [133]
Cw*02	axSpA	Moroccan [134]
C*15:02	axSpA	Korean [135]
C*15	AS	Chinese, Taiwanese, and Korean AS [53]
C*12	PsA	Canadian [125]
C*12:02:02	axSpA	Taiwanese [136]
Cw6	PsA	Argentinian [124]
C*06:02	PsA	Chinese [132]
F*01:01:01/01:03:01	AS	Portuguese [137]

*B*07* [55,116], *B*57:01* [54,55], *B*51:01* [55,116] have been proven negatively associated with AS.

**Table 5 ijms-25-06081-t005:** MHC class II allele associations.

MHC Class II	SpA Associations	Ancestry
DR4	AS	German [138]
DR1	AS	British [139]
	uSpA	Mexican [128]
DRB1*04	ReA	Tunisian [113]
	axSpA	Colombian [129]
DRB1*01	axSpA	Colombian [129]
DRB1*04:05	AS	Chinese Han [133]
DQB1*04	AS	Chinese, Taiwanese, and Korean [53]
DRB1*11, DPB1*03:01	AS	European, Asian, African [55]

DRB1*15:01, DQB1*02:01, and DQB1*06:02 have been proven to be negatively associated with AS [54,55].

**Table 6 ijms-25-06081-t006:** Species of Firmicutes Phylum in the composition of intestinal microbiota of SpA patients.

Class	Order	Family	Genus	Species
Clostridia	Clostridiales	Lachnospiraceae ▪ Abundance [145,146] ▪ Deficiency [147]	*Coprococcus* ▪ Abundance [145,146] ▪ Deficiency [148]	*C. comes* ▪ Deficiency [149]
			*Blautia* ▪ Abundance [146,150] ▪ Deficiency [151]	*B. pruducta* ▪ Abundance [146]
			*Roseburia* ▪ Abundance [145]	*R. insulinivorans* ▪ Deficiency [149]
			*Lachnospira* ▪ Deficiency [150,151]	
			*Dorea* ▪ Abundance [146,150]	
			*Pseudobutyrivibrio* ▪ Deficiency [148]	
		Eubacteriaceae	*Eubacterium*	*E. siraeum* ▪ Abundance [152] *E. ruminatum* ▪ Deficiency [151]
		Clostridiaceae	*Clostridium* ▪ Deficiency [147]	*C. bolteae*, *C. hathewayi* ▪ Abundance [149] *C. difficile* ▪ Deficiency [150]
		Ruminococcaceae ▪ Abundance [145,146]	*Ruminococcus* ▪ Abundance [146] ▪ Deficiency [148]	*R. gnavus* ▪ Abundance [146,147]
			*Faecalibacterium*	*F. prausnitzii* ▪ Deficiency [147]
Bacilli	Lactobacillales	Streptococcaceae	*Streptococcus* ▪ Abundance [147,151]	
		Lactobacillaceae	*Lactobacillus* ▪ Abundance [147]	
	Bacillales	Bacillaceae	*Bacillus*	
Negativicutes	Selenomonadales	Acidaminococcaceae	*Acidaminococcus*	*A.fermentans* ▪ Abundance [152]
		Veillonellaceae ▪ Deficiency [145]	*Megamonas* ▪ Abundance [150]	
			*Dialister* ▪ Abundance [151,153]	

**Table 7 ijms-25-06081-t007:** Species of Bacterioidetes Phylum in the composition of intestinal microbiota of SpA patients.

Class	Order	Family	Genus	Species
Bacteroidia ▪ Deficiency [147,154]	Bacteroidales	Rikenellaceae ▪ Abundance [145]		
		Porphyromonadaceae ▪ Abundance [145]	*Parabacteroides* ▪ Abundance [145]	*P. distasonis* ▪ Abundance [152]
		Bacteroidaceae ▪ Abundance [145]	*Bacteroides* ▪ Deficiency [147,151,155]	*B. coprophilus* ▪ Abundance [152]
		Prevotellaceae ▪ Deficiency [145,146]	*Prevotella* ▪ Abundance [151]	*P. copri* ▪ Abundance [152,155]
				*P. melaninogenica* ▪ Abundance [155]
				*Prevotella* sp. C561 ▪ Abundance [155]
			*Alloprevotella* ▪ Abundance [151]	

**Table 8 ijms-25-06081-t008:** Species of Proteo-, Actinobacteria, and Verrucomicrobia Phylum in the composition of intestinal microbiota of SpA patients.

Phylum	Class	Order	Family	Genus
Proteobacteria	Gamma-proteobacteria	Enterobacterales	Enterobacteriaceae ▪ Abundance [147,154]	*Shigella* ▪ Abundance [147]
				*Escherichia* ▪ Abundance [147]
				*Enterobacter* ▪ Deficiency [155]
				*Citrobacter* ▪ Deficiency [155]
	Beta-proteobacteria	Burkholderiales	Comamonadaceae	*Comamonas* ▪ Abundance [151]
		Neisseriales	Neisseriaceae	*Neisseria* ▪ Abundance [155]
Actinobacteria	Actinobacteria ▪ Abundance [147]	Actinomycetales	Actinomycetaceae	*Actinomyces* ▪ Abundance [146,155]
		Bifidobacteriales	Bifidobacteriaceae	*Bifidobacterium* ▪ Abundance [149,155]
		Coriobacteriales	Coriobacteriaceae ▪ Abundance [146]	*Collinsella* ▪ Abundance [151]
Verrucomicrobia	Verrucomicrobiae	Verrucomicrobiales	Akkermansiaceae	*Akkermansia* ▪ Deficiency [148]

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
