# Peer review of "Exploring the Pathogenesis of Spondylarthritis beyond HLA-B27: A Descriptive Review"

_ijms, 2024, doi:10.3390/ijms25116081_

Round 1

Reviewer 1 Report

Comments and Suggestions for Authors

This is a highly and well-referenced review article providing a timely and thorough update on the current understanding of the pathology of Spondyloarthritis.

1. The introduction would benefit from additional information on the prevalence and healthcare associated costs related to SpA to illustrate the importance and impact of this review.  Also, a more detailed clinical description of the symptoms and course of SpA is important information for a more general audience.

2. Again, for a more general audience, a few sentences describing HLA-B27 would be very helpful, including related basic immunology and connections to other inflammatory disease processes.

3. I did not see reference to prior review articles on SpA, and I think it’s important to cite these works in order to highlight the added value of this work.

4. A more robust conclusion that also considers remaining knowledge gaps and potentially important future directions of work on SpA would enhance the article.

Reviewer 2 Report

Comments and Suggestions for Authors

This is a very interesting paper on this field.

Author Response

Thank you very much for taking the time to review our manuscript. We greatly appreciate your positive feedback and are glad to hear that you found our work valuable. Your encouragement is truly motivating.

Reviewer 3 Report

Comments and Suggestions for Authors

The manuscript titled ‘Exploring the Pathogenesis of Spondylarthritis Beyond HLA-2 B27: A Descriptive Review' by Ruxandra-Elena Nagîț et al, details review, and it revealed that to identify new therapeutic targets, it is paramount to consider a holistic approach to understanding the 24-stage pathogenesis of SpA.

The review appears to be valid and the references are appropriate. However, there are some specific issues for publication in IJMS.

1.     It would be good to include more clinical and experimental in vivo studies on spondyloarthritis in introduction part.

2.     HLA-B27 accounts for 20.1%. Can it be said to be an important factor?

3.     In addition to HLA-B27, other factors were also added. Do you have a reference to what proportion these factors correspond to?

4.     Are there any cases where a treatment targeting HLA-B27 has been developed?

5.     It would also be good to include treatments that target other factors related to SpA.

6.     Please replace the reference with the latest one.

Comments on the Quality of English Language

Minor editing of English language required
